# Animal Models of Post-Traumatic Epilepsy

**DOI:** 10.3390/diagnostics10010004

**Published:** 2019-12-19

**Authors:** Kristin A. Keith, Jason H. Huang

**Affiliations:** 1Department of Neurosurgery, Neuroscience Institute, Baylor Scott and White Health, Central Division, Temple, TX 76508, USA; Kristin.Keith@BSWHealth.org; 2Department of Surgery, Texas A&M University Health Science Center, College of Medicine, Temple, TX 76504, USA

**Keywords:** post-traumatic epilepsy, traumatic brain injury, animal models, fluid percussion injury, controlled cortical impact model, weight drop model, pediatric traumatic brain injury, penetrating brain injury, acceleration impact model

## Abstract

Traumatic brain injury is the leading cause of morbidity and mortality worldwide, with the incidence of post-traumatic epilepsy increasing with the severity of the head injury. Post-traumatic epilepsy (PTE) is defined as a recurrent seizure disorder secondary to trauma to the brain and has been described as one of the most devastating complications associated with TBI (Traumatic Brain Injury). The goal of this review is to characterize current animal models of PTE and provide succinct protocols for the development of each of the currently available animal models. The development of translational and effective animal models for post-traumatic epilepsy is critical in both elucidating the underlying pathophysiology associated with PTE and providing efficacious clinical breakthroughs in the management of PTE.

## 1. Introduction

Traumatic Brain Injury (TBI) is the leading cause of morbidity and mortality, with an estimated annual incidence of 69 million individuals worldwide [1]. Post-traumatic epilepsy (PTE) is a recurrent seizure disorder, secondary to trauma to the brain tissue, and is one of the most devastating complications associated with traumatic brain injury, with the cumulative incidence of post-traumatic epilepsy ranging widely from 2% to >50% depending on the severity of the injury. Researchers have classified post-traumatic epilepsy into the following categories: (1) Immediate seizures, usually defined as those occurring within 24 h after initial injury, (2) early seizures, usually defined as those that occur less than one week after initial injury and (3) late seizures, usually defined as those that occur more than one week after initial injury [2,3]. It is well-established that the incidence of post-traumatic epilepsy increases with the severity of traumatic brain injury [2,4]. However, unfortunately, the mechanism by which traumatic brain injury leads to recurrent seizures is poorly understood.

Several risk factors have been suggested for the development of post-traumatic epilepsy, including: History of alcohol abuse (RR 2.18), post-traumatic amnesia (RR 1.31), focal neurologic deficit (RR 1.42), loss of consciousness at initial injury (RR 1.62), skull fracture (RR 2.27), midline shift on initial neuroimaging (RR 1.46), brain contusion (RR 2.35), subdural hematoma (RR 2.00) and intracranial hemorrhage (RR 2.65) [5,6,7,8]. Overall, patients with post-traumatic epilepsy pose a 2.53 fold increased risk of mortality as compared to traumatic brain injury patients without PTE [6]. To date, the current management of post-traumatic epilepsy remains prophylactic administration of phenytoin or levetiracetam within the first seven days following initial injury; however, studies have demonstrated that there is no evidence for effectiveness of any pharmacological therapy in the prevention or treatment of symptomatic seizures in post-traumatic epilepsy, and only low-level evidence was identified for non-pharmacologic therapy in significantly reducing seizures in post-traumatic epilepsy [9,10]. First-line therapy for the management of PTE remain agents that have shown efficacy in focal epilepsy, and include levetiracetam or oxcarbazepine, although, not uncommonly, post-traumatic epilepsy is refractory to medical management [10,11,12,13]. Reliable animal models are essential to elucidate the pathophysiology of post-traumatic epilepsy and further evaluate potential treatment options. This review will focus on the current various animal models available in translational research of post-traumatic epilepsy.

## 2. Animal Models of Post-Traumatic Epilepsy

### 2.1. Fluid Percussion Injury

Fluid Percussion Injury remains the most extensively utilized and highly studied model of post-traumatic epilepsy that has successfully demonstrated an increase in seizure susceptibility, as measured by a second-hit chemoconvulsant challenge, as well as the production of spontaneous epileptiform discharges, the hallmark of epilepsy [14,15]. (Table 1) Traditionally, the Fluid Percussion Model was isolated to rats; however, recent studies have successfully replicated an increased seizure susceptibility in mice at 30 days post-fluid percussion injury [14], as well as 6-month post-fluid percussion injury [16,17].

#### 2.1.1. Protocols for Developing Fluid Percussion Model

Animals are anesthetized as per institutional protocol—then, their heads of the animals are shaved, and inserted into a stereotaxic frame. Strict sterile technique is maintained throughout the course of surgical procedure as detailed below.

##### Protocol for Fluid Percussion Model in Mice

A 3 mm craniotomy was performed with dura intact over the right parietal cortex between the lambda and the bregma approximately 2 mm to the right of midline. A 3 mm injury cap (made from a female leur-lock cap) is positioned over the craniectomy site and secured with glue. A small amount of methyl-methacrylate dental acrylic solution is used to create a seal with cement around the injury hub to ensure the fluid bolus remains within the cranial cavity. The injury hub is filled with 0.9% NaCl to keep the dura moist during the recovery phase. Injury pressure pulses are delivered ranging 0.9–2.1 atm in magnitude and lasting approximately 20 ms in duration over the exposure dura [18]. Injury cap was removed, scalp sutured, and animals returned to home cages for recovery.

##### Protocol for Fluid Percussion Model in Rats

Initially described by McIntosh et al., FPI model was produced with a 4.8 mm craniectomy centered over the left parietal cortex mid-distance between the bregma and lambda and mid-distance between the sagittal suture and lateral ridge (4.5 mm Anteroposterior, 3.0 mm Medio-lateral) [19]. The Fluid Percussion Injury model has since been further modified in rats with a 3 mm craniectomy performed centered 3 mm lateral from bregma and 3.5 mm to the left of the sagittal suture [20]. A 3 mm injury cap (made from a female leur-lock cap) is positioned over the craniectomy site and secured with glue. A small amount of methyl-methacrylate dental acrylic solution is used to create a seal with cement around the injury hub to ensure the fluid bolus remains within the cranial cavity. The injury hub is filled with 0.9% NaCl to keep the dura moist during the recovery phase. Injury pressure pulses are delivered depending on injury severity, with low magnitude injury at 1.5 atm pressure pulse lasting 20 ms in duration and high magnitude injury at 3–3.4 atm pressure pulse lasting 20 ms in duration [21]. Injury cap was removed, scalp sutured, and animals returned to home cages for recovery.

#### 2.1.2. Evidence of Seizure Susceptibility Following Fluid Percussion Model

Fluid Percussion Injury (FPI) model has been shown to successfully reproduce histopathology associated with traumatic brain injury, including a focal contusion within the cerebral cortex with accompanying petechial or intraparenchymal hemorrhage, cerebral edema and progressive gray matter damage. Additionally, following FPI, the corneal, pinnal, paw flexion and righting responses are transiently lost and spontaneously rapidly return, as seen in the acute phase of traumatic brain injury patients with transient loss of reflexes—suggesting additional validity of the FPI model [22].

To evaluate an increase in seizure susceptibility following induction of the fluid percussion injury (FPI) model, a Pentylenetetrazol (PTZ) test is frequently performed. Data following a single bolus administration of a subconvulsant dose of PTZ demonstrated a statistically significant increase in the occurrence of PTZ-induced seizures in FPI model vs. control animals, as well as the overall increased mortality in FPI model vs. control animals. Furthermore, enhanced seizure susceptibility to PTZ-induced seizures persisted for at least six months following induction of FPI, even in animals that had no evidence of behavioral spontaneous seizures [23].

Shultz et al., 2013, suggested that subtle changes in the ipsilateral hippocampus in the acute phase following induction of FPI may be related to the development of PTE. Specifically, it was found that significant changes in the ipsilateral hippocampal surface shape exist in rats that go on to develop epilepsy just one-week status post FPI as compared to control models, and additionally, F-FDG PET parameters from the ipsilateral hippocampus were able to accurately predict the epileptic outcome in such models [24].

#### 2.1.3. The Validity of Fluid Percussion Model

FPI model demonstrates increased seizures susceptibility and reproduces the histopathology associated with traumatic brain injury, including, diffuse white matter injury of varying severity, a focal contusion within the cerebral cortex with accompanying petechial or intraparenchymal hemorrhage, cerebral edema, progressive gray matter damage and white matter tissue tears suggesting a high construct validity [22,25]. Furthermore, FPI model demonstrates persistent neuromotor and cognitive deficits up to one year following severe FPI with the associated temporal pattern of cellular death that recapitulate what is seen in traumatic brain injury patient [26,27]. In conclusion, as an experimental model of traumatic brain injury and post-traumatic epilepsy, FPI model provides consistency, reproducibility and reliability required as a laboratory model with overall accepted construct validity. Unfortunately, due to the mechanism by which the FPI model is produced, the therapeutic predictive validity appears to be lacking, and experimentation regarding therapeutic interventions have yet to accurately translate to clinical efficacy in the management of both traumatic brain injury and post-traumatic epilepsy.

### 2.2. Controlled Cortical Impact Injury

Controlled cortical impact (CCI) model produces brain injury by using a pneumatic or electromagnetic impactor to compress exposed brain to cause varying severity of brain injury [28]. CCI has been successfully shown to mimic injuries produced with traumatic brain injury, including cortical tissue loss, acute subdural hematoma, axonal injury, concussion and blood-brain barrier (BBB) dysfunction [29]. To date, CCI models have been successfully replicated in ferrets, rats, mice, swine and monkeys [17]. Initially thought to produce focal gray matter injury, studies have suggested a more widespread degeneration, including white matter, hippocampus and thalamus [30].

#### 2.2.1. Protocols for Developing Controlled Cortical Impact (CCI) Model

Animals are anesthetized as per institutional protocol, (as previously mentioned). Strict sterile technique is maintained throughout the course of surgical procedure. A midline incision is made, soft-tissues reflected and a craniectomy performed as detailed below.

##### Protocol for Controlled Cortical Impact (CCI) Model in Mouse

A 5 mm craniotomy performed over the left parietotemporal cortex, while maintaining dura mater intact. The controlled cortical impact is conducted using a beveled 3 mm flat-tip impounder at velocity 5.7–6 m/s with cortical deformation 1 mm [31]. Surgical site closed with suture.

##### Protocol for Controlled Cortical Impact (CCI) Model in Rats

10 mm craniotomy performed centrally between the bregma and lambda, while maintaining the dura mater intact. Low (6 m/s) with cortical deformation 1 mm, Medium (6 m/s) with cortical deformation 2 mm and High (6 m/s) with cortical deformation 3 mm produced increasing injury severity with an associated increase in histopathology seen both grossly and microscopically [28]. Surgical site closed with suture.

##### Protocol for Controlled Cortical Impact (CCI) Model in Ferrets

A 1.5 cm diameter craniotomy performed on the midline, centered between bregma and lambda, while maintaining dura mater intact. Bone flap stored in saline during the remainder of surgical procedure. Controlled cortical impact performed on the midline to intact dura mater overlying the lateral cerebral gyri, posterior to the cruciate sulcus and medial to the lateral sulci with variation in velocity and impact depth depending on the desired degree of injury. Low velocity (1.9 m/s) with cortical deformation 3.5 mm, Medium Velocity (2.86 m/s) with cortical deformation 3.0–4.5 mm and High velocity (4.0 m/s) with cortical deformation 2.5–4.0 mm produced increasing injury severity with an associated increase in histopathology seen both grossly and microscopically [32]. Surgical site closed with suture.

#### 2.2.2. Evidence of Seizure Susceptibility Following Controlled Cortical Impact (CCI) Model

It has been demonstrated in recent studies that although a large proportion of CCI mice do not develop spontaneous seizures, spontaneous epileptiform spiking occurs suggestive of ongoing epileptogenesis, with an overall increase in seizure susceptibility following injury with an associated increased mortality rate. This was further assessed and demonstrated shortened latency to the first epileptic discharge in 60% of CCI mice, increased total number of epileptic discharge spikes (at least one standard deviation above the mean) within the first 60 min following administration of PTZ in CCI mice and a increased number of total seizures (at least one standard deviation above the mean) following administration of PTZ in CCI mice [16].

Both early post-traumatic seizures (within 24 h following initial head trauma) and delayed post-traumatic seizures up to 71 days have been successfully reproduced following severe CCI [33]. Hunt et al., 2009, established a comparable incidence of unprovoked seizures weeks following controlled cortical impact injury as compared to clinical studies, with 20–36% CCI model vs. 39% patients sustaining severe TBI with intact dura [33].

Yang et al. established progressive hyperexcitability in neocortical circuits within the subacute phases (within the first two weeks) following severe CCI, manifested as the development of prolonged evoked burst discharges and spontaneous epileptiform activities [30]. Unlike other animal models of traumatic brain injury, CCI models result in several chronic alterations, including mossy fiber sprouting and delayed hippocampal lesions—classically defined as a hallmark for temporal lobe epilepsy [30,34,35].

#### 2.2.3. The Validity of the Controlled Cortical Impact (CCI) Model

CCI model has increased seizure susceptibility, and previous studies have demonstrated recapitulation of histopathology particularly associated with closed head trauma. Histopathological analysis reveals a large cavitary lesion, developing 6–24 h post-injury with surrounding intense staining of the adjacent cortex, hippocampus and ipsilateral dorsolateral thalamus [34]. Staining was noted to reach a maximum at 48 h post-injury with extension laterally into the auditory cortex, anteriorly into the dorsal striatum and posteriorly into the visual cortex [35]. Furthermore, results have indicated that following severe CCI, progressive development of neocortical hyperexcitability occurs, ultimately leading to spontaneous epileptiform discharges suggestive of a rapid epileptogenic process [30].

In conclusion, as an experimental model of traumatic brain injury and post-traumatic epilepsy, CCI model, similarly to FPI model, provides consistency, reproducibility and reliability required as a laboratory model with an overall accepted construct validity. CCI model appears to have the additional benefit over FPI model with the ease of altering the severity of the injury and the development of both unprovoked seizures and hippocampal pathology at an earlier timepoint in comparison (CCI: Ten weeks post-injury vs. FPI: Months post-injury) [33] making quality experimentation more feasible.

### 2.3. Impact-Acceleration Model

Impact-Acceleration Model, also known as the Weight Drop Model, mimics diffuse traumatic brain injury. While this model has a comparatively easier protocol with inexpensive supplies, it fails to produce post-traumatic epilepsy at intensities that are not frequently lethal [36].

#### 2.3.1. Protocols for Developing Impact-Acceleration Model

Animals are anesthetized as per institutional protocol—then, their heads of the animals are shaved, and inserted into a stereotaxic frame. Strict sterile technique is maintained throughout the course of surgical procedure. A 3 mm craniotomy was performed centered 2.3 mm caudal and 2.3 mm lateral to bregma. A weight-drop device was placed on the stereotactic arm over the dura and adjusted to 2.5 mm below the dura. 20 g weight was dropped from 20 cm above the exposed, intact dura mater. Scalp sutured closed, and animals returned to housing for recovery [23,37].

Furthermore, previous studies have indicated that graduations of mechanical impact level reproduces progressing degrees of injury severity, and thus, varying degrees of severity of traumatic brain injury models, simply by altering the heights by which the weights are dropped [38].

#### 2.3.2. Evidence for Seizure Susceptibility in Impact Acceleration Model of Post-Traumatic Epilepsy

During the acute phase, mild-to-moderate convulsions were observed, with late convulsions incidentally discovered in impact-acceleration models lasting up to 15 weeks post-injury. Following a single sub-convulsive dose of PTZ, impact-acceleration model demonstrated an increase in both frequency and severity, with 62.5% impact-acceleration models demonstrating Class V seizures and approximately 60% of those that developed Class V seizures having repeated episodes. Furthermore, the median convulsion category in impact-acceleration model was generalized tonic-clonic (Class IV) as compared to Class I in control rats [23].

#### 2.3.3. The Validity of Impact-Acceleration Model of Post-Traumatic Epilepsy

Impact-Acceleration model has successfully induced a reproducible lesion localized to the somatosensory cortex that undergoes massive atrophy during the immediate two-week period following initial injury resulting in maximal size lesion of approximately 30 cubic mm with associated selective ipsilateral hippocampus damage [23]. Although the impact-acceleration model has successfully demonstrated increased susceptibility to PTZ-evoked seizures, unlike other models of post-traumatic epilepsy, impact-acceleration model fails to demonstrate spontaneous seizures—a hallmark of post-traumatic epilepsy. Furthermore, as mentioned previously, the impact-acceleration model fails to consistently recapitulate post-traumatic epilepsy without using intensities that frequently cause mortality.

Until recently, in comparison to the other animal models discussed in this review, the impact-acceleration model as initially described by Marmarou et al. lacks precision or ease of reproducibility in results typically desired in a well-studied animal model [17,38,39,40]. This, however, was disproved following Hsieh et al., 2017, who went on to demonstrate that by controlling impact height and pressure, it is possible to reliably induce injury of graded neurologic severity with a positive correlation with behavioral tests [38].

### 2.4. Canine Model of Post-Traumatic Epilepsy

In the 1980s, several large animal models of traumatic brain injury were well described, including the induction of traumatic brain injury in both canines and felines; however, until recently, have not been assessed as a resource for understanding post-traumatic epilepsy [41].

#### 2.4.1. Evidence for Seizure Susceptibility in Canine Model of Post-Traumatic Epilepsy

Steinmetz et al., 2013, performed a retrospective analysis of the natural history of canine head trauma to determine the risk of developing post-traumatic epilepsy following head trauma in canines and demonstrated that 18.6% of canines that experienced head trauma developed early and/or late-onset seizures. Of the canines that experienced head trauma, 14% developed early-onset seizures, particularly within the first 24 h following initial injury; however, as seen in the human traumatic brain injury population, there was an increased risk of post-traumatic epilepsy correlating with increased severity of the head injury. This was further supported with the notion that of the canines with penetrating injuries, 44% developed early-onset seizures as compared to the 12.8% seen in closed head injuries [42].

#### 2.4.2. The Validity of the Canine Model of Post-Traumatic Epilepsy

Currently, studies involving traumatic brain injury in canine models are limited and lack evaluation of the development of post-traumatic epilepsy. Although Steinmetz et al., 2013, supported the notion that head trauma, particularly involving severe TBI resulting from skull fracture, correlates with a significantly increased risk of developing epilepsy, additional studies would be required for validation of the canine model of post-traumatic epilepsy with particular focus in developing canine animal models or evaluation of the previously discussed TBI canine animal models [41,42].

### 2.5. Model of Post-Traumatic Epilepsy after Penetrating Brain Injuries

In the spectrum of Traumatic Brain Injury, penetrating brain injuries have been shown to have a significantly elevated risk of post-traumatic epilepsy. Gunshot wounds to the head are one of the most common sources of penetrating brain injuries and commonly have retained fragments, including bone fragments, bullets or bullet fragments—which are predominantly composed of copper-coated lead [1,43].

#### 2.5.1. Protocols for Developing Penetrating Brain Injury Model of Post-Traumatic Epilepsy

Animals are anesthetized as per institutional protocol—then, their heads of the animals are shaved, and inserted into a stereotaxic frame. Strict sterile technique is maintained throughout the course of surgical procedure. Skull was exposed, and high-speed drill was mounted vertically on the stereotactic arm. Hardened steel burr (1.5 mm diameter) was introduced through the skull, dura and brain at 1000 RPMs using the stereotactic arm (Anteroposterior: −4.9 mm, Mediolateral: 4.5 mm to the bregma) to a depth of 8 mm below the brain surface. Copper wire (0.02 in diameter, 5 mm in length) or stainless-steel wire (0.03 in diameter, 5 mm in length) was introduced into the lesion immediately after it had been created and scalps were closed [16].

#### 2.5.2. Evidence for Seizure Susceptibility in Penetrating Brain Injury Model of Post-Traumatic Epilepsy

Ninety-six percent of the animal models with embedded copper-wire developed chronic epilepsy, with an increase in both seizure frequency (>3 seizures per day vs. 0.2 seizures per day) and the seizure severity (Racine score 4 vs. Racine score 1) as compared to both control animals and stainless-steel embedded animals. Furthermore, rats with embedded copper-wire demonstrated the enlargement of the lesion to approximately 3–4 mm mediolateral and 5–6 mm anteroposterior with associated necrosis at seven months post-injury [44]. Of the penetrating brain injury models, only Kendirli et al. has been successful in demonstrating spontaneous seizures [44].

### 2.6. Pediatric Post-Traumatic Epilepsy

Post-traumatic seizures affect 12–35% of children following traumatic brain injury and are associated with a significantly worse cognitive and functional outcome regardless of the severity of injury [45,46,47]. Furthermore, suppressive and prophylactic therapy for post-traumatic epilepsy is often largely ineffective in the pediatric population, with nearly 60% of cases being intractable to medical management [3,48,49].

The investigation regarding the management of pediatric post-traumatic epilepsy remains limited, and the pathogenesis remains poorly understood. Until recently, clinically relevant models for post-traumatic epilepsy in the pediatric population were lacking; however, previous studies have worked to develop pre-clinical models of pediatric PTE with some success [50].

#### 2.6.1. Protocols for Developing Pediatric Post-Traumatic Epilepsy Models

Litters of Sprague-Dawley Rats were housed with lactating females until weaning on the post-natal day (PND) 21, at which time they were segregated and housed in a temperature and light-controlled environments. Animals are anesthetized as per institutional protocol—then, their heads of the animals are shaved and inserted into a stereotaxic frame. A 6 mm × 6 mm craniotomy was performed over the left parietal cortex centered around 4 mm anterior, 4 mm lateral to the bregma. Rats underwent controlled cortical impact (CCI) as per protocol centered over the left parietal cortex 4 mm rostral to lambda and 4 mm left of midline with a 5 mm round-tip at 4 m/s velocity with 2 mm deformation and 100 ms duration) on PND 16–18 [51].

#### 2.6.2. Evidence of Seizure Susceptibility Following Pediatric Post-Traumatic Epilepsy Model

Statler et al., 2009, performed a series of three separate standard electroconvulsive seizure paradigms to assess hindbrain, forebrain and limbic seizure threshold in both adolescence (defined as PND 34–40) and maturity (defined as PND 60–63) in Sprague-Dawley rats. Tonic Hindlimb Extension (THE) seizures were used to assess hindbrain seizure thresholds, Minimal Clonic seizures were used to assess forebrain seizure thresholds, and Partial Psychomotor seizures were used to assess limbic seizure thresholds. Overall, CCI to the left parietal cortex during immaturity appeared to lower minimal clonic seizure threshold at maturity and partial psychomotor seizure threshold during adolescence. Of note, THE seizure threshold was lower in both sham-surgery rats and TBI rats, suggesting that anesthetic administration during immaturity alone may lower the seizure threshold and provide an additional limitation with experimental models for translational research in pediatric traumatic brain injury [45,51].

Additionally, Statler et al., 2008, demonstrated that MRI scans obtained in post-CCI rats recapitulate the histopathology associated with traumatic brain injuries, including cavitary lesions, cortical surface damage and hippocampal loss ipsilateral to the controlled cortical impact corresponding to approximately 30% of total brain volume. A later study performed by Statler et al., demonstrated no evidence of EEG spiking or seizures in controls rats, although successfully demonstrated EEG spiking in 87.5% post-CCI rats and in rare occasions resulted in spontaneous, recurrent seizures, proposing that CCI performed during immaturity may produce an accurate representation of the pediatric traumatic brain injury model [45,51].

#### 2.6.3. The Validity of Pediatric Post-Traumatic Epilepsy Model

Due to limited studies involving the proposed pediatric model of post-traumatic epilepsy, it is difficult to ascertain the construct validity of the model as a whole. The findings presented by Statler et al. are suggestive of a promising model to recapitulate the histopathology and increased risk of seizure threshold seen in the clinical setting of pediatric traumatic brain injury, however, remains inconclusive and ultimately, requires further investigation [51].

## 3. Conclusions

Undoubtedly, the variation in clinical presentation, degree of latency and severity of traumatic brain injury poses a challenge in the diagnosis and management of post-traumatic epilepsy. Translation of successful in vitro research studies into clinical therapeutics is a major challenge with the potential for costly and prolonged clinical trials. The present review outlines the currently available animal models in post-traumatic epilepsy and succinct protocols used in developing each. Despite the important advances in both traumatic brain injury and post-traumatic epilepsy, clinical evidence for the management of PTE remains elusive. Although the current animal models of post-traumatic epilepsy provide key evidence in understanding post-traumatic epilepsy, no current model to date fully recapitulates traumatic brain injury, as shown in Table 1. Therefore, understanding the strengths and weaknesses of each animal model may be key in the determination of which animal model provides the necessary requirements for the evaluation of specific research questions.

## Figures and Tables

**Table 1 diagnostics-10-00004-t001:** Advantages and Disadvantages of Current Available Animal Models of Post-Traumatic Epilepsy.

Animal Model	Advantages	Disadvantages
Fluid Percussion Injury (FPI)	- Recapitulates histopathology associated with Traumatic Brain Injury including: diffuse white matter injury, focal contusion, cerebral edema, progressive gray matter damage - Increased seizure susceptibility at both 30 days and 6-months post-injury - Demonstrates persistent neuromotor and cognitive deficits up to 1-year post-injury	- Prolonged time course to development of increased seizure susceptibility (i.e., Months post-injury) - Lacks translation to therapeutic predictive validity
Controlled Cortical Impact	- Recapitulates histopathology associated with Traumatic Brain Injury including: cortical tissue loss, acute subdural hematoma, axonal injury, concussion, blood-brain barrier dysfunction - Shorter time course to development of increased seizure susceptibility (i.e., Weeks post-injury) - Development unprovoked seizures and chronic findings defined as hallmark associated with temporal lobe epilepsy (i.e., mossy fiber sprouting, delayed hippocampal lesions)	- Complex technical device required for production of CCI injury - Mechanical Variation - Unable to recapitulate the full breadth of Traumatic Brain Injury seen clinically (i.e., limited diffuse effects related to CCI animal models as compared to other PTE animal models)
Impact Acceleration Model	- Simplistic protocol - Inexpensive - Ease in producing progressive degrees of injury severity	- Difficulty with reliable reproducibility - High mortality rate associated with impact levels required for development of histopathology and/or increased seizure susceptibility associated with TBI and development of post-traumatic epilepsy - Failure to consistently produce spontaneous seizures
Canine Model of Post-Traumatic Epilepsy	- Natural history of head trauma in canines serving as potential model for post-traumatic epilepsy	- Lack of evidence or evaluation regarding production of canine post-traumatic epilepsy model
Penetrating Head Trauma Model	- Copper-embedded model shown to have significantly increased seizures susceptibility and increased mortality rate	- Lack of evidence regarding copper-embedded wire truly recapitulating penetrating head trauma - Limited efficacy in producing spontaneous seizures in currently available models for penetrating head trauma
Pediatric Post-Traumatic Epilepsy	- CCI injury during immaturity appears to lower seizure threshold during maturity and adolescence	- Lack of evidence and limited evaluation regarding validity of pediatric post-traumatic epilepsy model due to isolated study

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
