# Peer review of "Animal Models of Post-Traumatic Epilepsy"

_diagnostics, 2019, doi:10.3390/diagnostics10010004_

Round 1
Reviewer 1 Report
The manuscript entitled “Animal Model of Post-Traumatic Epilepsy” presents an interesting overview of the available animals' models for investigating post-traumatic epilepsy. The article is well written and provides interesting information in the field of experimental pharmacology.
I have some minor comments for the authors.
When you describe the protocols for developing fluid percussion model it good to add a subtitle that shows the types of models – rodents and also to add non-rodents model if exist.
For the protocols for developing impact-acceleration model please specify the species of animals used. The same observation for all the other models described, please indicate the species used.
It will be good to add a schematic representation of the advantages and disadvantages of each model used.
Author Response
To Whom it May Concern:
Thank you so much for your time and consideration with reviewing our article regarding Animal Models in Post-Traumatic Epilepsy!
For the various animal models, I have provided sub-categories for the various species involved in each of the animal models available. I have provided a schematic listing the main advantages and disadvantages associated with each of the given animal models.Thank you again for taking the time to review our article and provide feedback!
Reviewer 2 Report
This review of animal models of post traumatic epilepsy appears useful and adequately researched. One minor point in the abstract, I would not characterize post-traumatic epilepsy as being the most common complication associated with traumatic brain injury. Within the spectrum of common sequelae of traumatic brain injury, epilepsy although important is most likely not the most common.
Author Response
Thank you for your time and consideration in reviewing our article regarding Animal Models in Post-Traumatic Epilepsy!
The abstract has been adjusted regarding the statement that previously characterized post-traumatic epilepsy as the most common complication of traumatic brain injury to state that there is an increasing incidence of post-traumatic epilepsy corresponding to the severity of traumatic brain injury.Thank you again for taking the time to review our article and provide feedback!
Sincerely,
Kristin Keith